# In Defense of Prompt-based Continual Learning: Task Interference Mitigation via Confidence-Stratified Classifier Calibration

## Abstract

Prompt-based Continual Learning is a promising direction that effectively leverages the capabilities of pre-trained models. Prompt-based methods typically learn task-specific prompts and predict task-ID to select the prompt during inference. However, task interference caused by prompt misselection significantly constrains their performance. Although several studies have attempted to mitigate this problem, they treat all samples equally, without specifically addressing the samples that are primarily responsible for this issue. In this paper, we first partition samples into three distinct categories according to their different responses to prompt misselection, including susceptible samples, refractory samples, and resilient samples. Based on such stratification, we unravel that susceptible samples are the main source of such task interference, as only their classification results are influenced by prompt misselection, while other samples remain unaffected. This realization drives us to design a novel prompt-based approach called **Co**nfidence-**S**tratified **C**lassifier Calibration (CoSC), to mitigate task interference arising from prompt misselection by targeting the root cause. Specifically, we leverage the unique properties of each sample category to calibrate classifiers of both the prompt instruction part and the prompt selector, thereby reducing the exposure of susceptible samples to incorrect prompts. Extensive experiments show that CoSC outperforms prompt-based counterparts and achieves state-of-the-art performance across various benchmarks under class-incremental setting.

## 1 Introduction

Continual learning (CL) aims to develop models that are able to learn from a stream of data over time without forgetting previous knowledge. Consequently, CL faces a key challenge in task interference (Nori & Kim, 2024; Riemer et al., 2018; Aljundi et al., 2019; Kanakis et al., 2020; Wang et al., 2023b) where new and old tasks interfere with each other, leading to the stability-plasticity dilemma (Mermillod et al., 2013; Wu et al., 2021; Jung et al., 2023; Abraham & Robins, 2005). Specifically, when learning new tasks, the interference disrupts previously acquired knowledge, damaging the model's stability. Conversely, old task knowledge can hinder the learning of new tasks, affecting the model's plasticity. The goal of CL is to maintain a balance between stability and plasticity, ensuring that the model can retain old knowledge while effectively learning new information. In this paper, we focus on the exemplar-free setting, where no samples from previous tasks are stored, making the mitigation of task interference more challenging.

In recent years, prompt-based continual learning methods have emerged as a promising approach to address the aforementioned challenge. The typical stream of prompt-based methods contains two parts (Wang et al., 2022b;a; Khan et al., 2023; Wang et al., 2023a; Li et al., 2024): (a) prompt instruction part that learns task-specific prompts, thus acquiring task-specific knowledge to guide the pre-trained model. (b) prompt selector that learns to predict task-ID, thus selecting an appropriate prompt for each sample during testing. Such framework partly mitigates task interference by isolating parameters, but task interference brought by prompt misselection is still severe. Consequently, existing methods depend heavily on precise prompt selection—a requirement that is difficult to satisfy. Although several studies have attempted to address this problem under such framework (Wang et al.,

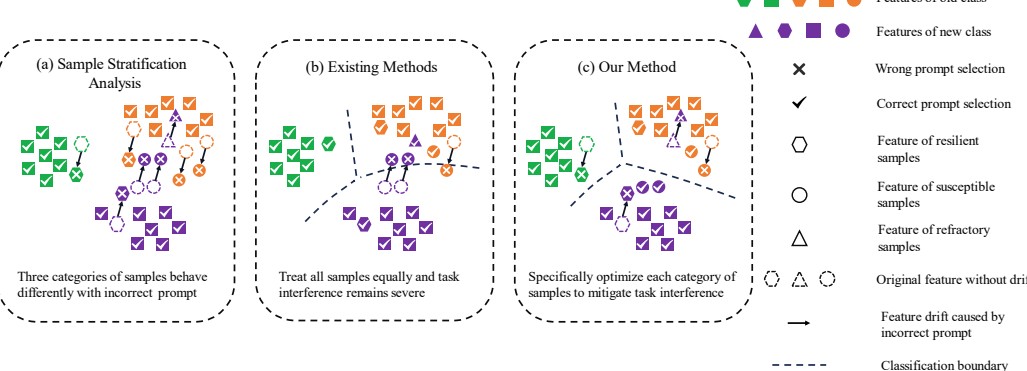

Figure 1: (a) Different samples exhibit varying responses to prompt misselection, which allows them to be stratified into three distinct categories. Among them, only susceptible samples have their classification outcomes affected by prompt misselection, while the other two types of samples remain unaffected. (b) Existing methods treat all samples uniformly, leading to frequent misclassification of susceptible samples due to prompt misselection. (c) Our method specifically tailors learning to each sample category, effectively mitigating task interference.

2023a; Gao et al., 2024; Le et al., 2024; Li et al., 2024), they have not thoroughly analyzed the reasons underlying existing methods' reliance on precise prompt selection, leading to one-sided solutions.

To more effectively mitigate this issue, we examined the effects that prompt misselection imposes on each sample. As illustrated in Figure 1(a), individual samples exhibit heterogeneous responses to prompt misselection, enabling us to stratify them into three distinct categories: (1) Susceptible samples: Only these samples can have their predictions flipped from correct to wrong with an incorrect prompt. Hence, these samples constitute the primary factor that constrains model performance under prompt misselection. (2) Refractory samples: These samples consistently yield wrong predictions, irrespective of whether an erroneous prompt is employed. (3) Resilient samples: These samples remain correctly classified regardless of which prompt is used. Therefore, reducing the incidence of prompt misselection for susceptible samples is key to mitigating task interference in prompt-based approaches. However, existing methods treat all samples uniformly, leaving a substantial number of susceptible samples misclassified as a result of their prompt misselection, as shown in Figure 1(b). Moreover, since prompt-based continual learning methods are designed for the exemplar-free setting, it's impossible to directly optimize the prompt selection of all susceptible samples due to the lack of old task data, which makes the problem still challenging.

Motivated by these observations, we propose an innovative approach called **Co**nfidence-**S**tratified **C**lassifier Calibration (CoSC) to mitigate task interference caused by prompt misselection of susceptible samples, as shown in Figure 1(c). Based on the discussed stratification of samples, we adopt a two-step approach to perform Confidence-aware Classifier Alignment (CCA) and then Loss Density-modulated Prompt Selector (LDS), thereby reducing the exposure of susceptible samples to misselected prompts. Specifically, we distinguish the three types of samples based on the confidence of the prompt instruction part, and then handle each type with targeted processing accordingly. For refractory samples, we drop them during classifier alignment to convert susceptible samples into resilient ones and use them as anchor points to the prompt selector, defending against classifier over-confidence. For susceptible samples, we design a piecewise weighting function to refine the cross-task decision boundary in a constrained manner and introduce a loss density-modulated mechanism to ensure their appropriate prompt selection. Additionally, the abundant resilient samples are employed to align classifiers with their task-representative knowledge and defend against overfitting of the prompt selector.

Our contributions include: (1) We first propose and experimentally validate that, in prompt-based continual learning, prompt misselection has markedly different impacts on different types of samples. (2) We stratify samples into three types based on their distinct responses to prompt misselection. Building on this stratification, we propose Confidence-aware Classifier Alignment and Loss Density-modulated Prompt Selector, leveraging the unique properties of each sample category to effectively mitigate task interference in prompt-based continual learning. (3) Our extensive experimental

results under class-incremental setting across multiple benchmarks convincingly demonstrate the effectiveness and robustness of our proposed method.

## 2  RELATED WORK

**Class Incremental Learning:** Class-incremental learning (CIL) sequentially learns new categories and tests without task identity information. Rehearsal based methods (Rebuffi et al., 2017; Chaudhry et al., 2019; Castro et al., 2018; Hou et al., 2019; Smith et al., 2024) maintain a small buffer of exemplars from previous tasks and replay them to mitigate catastrophic forgetting. Regularization-based (Kirkpatrick et al., 2017; Li & Hoiem, 2017; Zenke et al., 2017; Hou et al., 2018) approaches introduce additional loss terms, such as knowledge distillation, to preserve parameters or outputs crucial for previously learned classes, thereby reducing forgetting without storing raw data. Architecture based techniques (Rusu et al., 2016; Yoon et al., 2017; Mallya & Lazebnik, 2018; Yan et al., 2021) dynamically expand or isolate parts of the network. By adding neurons, modules, or freezing subsets of weights, they allocate dedicated capacity for incoming classes while preserving the knowledge of old classes. Moreover, some hybrid approaches (Chaudhry et al., 2019; Douillard et al., 2020) that combine replay, regularization, and architectural adaptation achieve a balance between memory efficiency and adaptability.

**Prompt-based Continual Learning:** Prompt-based continual learning has recently gained significant attention. The main stream of methods learns task-specific prompts while keeping previously learned prompts frozen (Wang et al., 2022b; 2023a; 2022a; Khan et al., 2023). However, these methods suffer severely from task interference brought by prompt misselection. Under such framework, some efforts have been made to mitigate this issue by improving the overall task-ID prediction accuracy (Li et al., 2024; Gao et al., 2024; Wang et al., 2023a; Khan et al., 2023). Nevertheless, all these methods treat all samples uniformly without realizing that only a subset of samples are susceptible to prompt misselection, leading to non-targeted optimization. Besides, RCS-Prompt (Yang et al., 2024) mitigates task interference by relabeling new class samples to old classes, inevitably sacrificing the model's performance on new classes. In this paper, we stratify samples into three categories based on their responses to prompt misselection to identify the reasons why existing methods suffer from prompt misselection. Building on this, we specifically leverage each category to address task interference at its core.

## 3  PRELIMINARIES

**Problem Formulation.** In Class-Incremental Learning (CIL), a model is trained on a sequence of $T$ disjoint tasks while needing to preserve previously learned knowledge. While learning task $t$, the model has access only to the current task's dataset, $\mathcal{D}_t = \{(x_i^t, y_i^t)\}_{i=1}^{N_t}$, where $x_i^t$ is an input image, $y_i^t$ is its corresponding label, and $N_t$ denotes the number of samples in task $t$. Each label $y_i^t$ belongs to the task's unique label set $\mathcal{Y}_t$, which is disjoint from all previously seen sets (i.e., $\mathcal{Y}_t \cap \mathcal{Y}_s = \emptyset$ for $s \neq t$). After training on task $t$, the model is evaluated on its ability to make predictions over the cumulative set of all classes learned so far, $\mathcal{Y}_{1:t} = \bigcup_{s=1}^{t} \mathcal{Y}_s$. The problem is made particularly challenging because, at test time, the task identity of an input sample is not provided, forcing the model to discriminate between all classes simultaneously.

**Prompt-based Continual Learning.** Prompt-based learning is a parameter-efficient paradigm for continual learning that adapts a large, frozen pre-trained backbone, denoted as $f_\theta$, to a sequence of downstream tasks. For each learning session on task $t$, a small set of trainable parameters, known as a task-specific prompt ($P_t \in \mathbb{R}^{L_p \times D}$), is learned, where $L_p$ is the prompt length and $D$ is the hidden size. These prompt tokens are prepended to the input feature tokens and subsequently adjust the features through the model's cross-attention mechanism. During training, only the current prompt $P_t$ is updated, while all previously learned prompts ($P_1, \ldots, P_{t-1}$) are kept frozen to preserve knowledge of old tasks. A core challenge arises at test time because the task ID is unavailable. To address this, the model must first predict the task identity of the input sample. It then selects the corresponding task-specific prompt associated with the predicted task to process the input and make a final prediction.

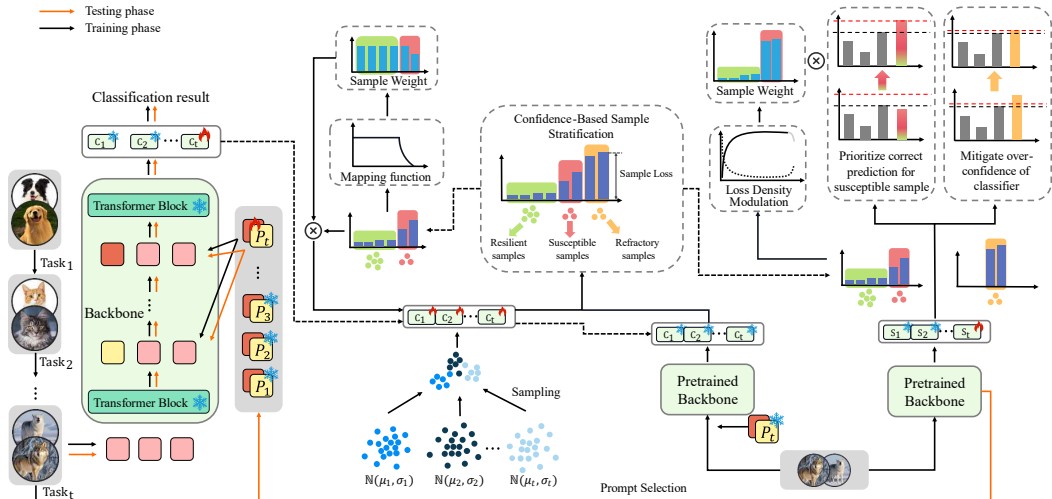

Figure 2: The framework of our proposed method. Our approach adopts a two-stage learning after learning task-specific prompt. Samples are first stratified into three categories based on their classification loss, reflecting their response to prompt misselection. Then, tailored strategies are applied to each category to develop our Confidence-aware Classifier Alignment (CCA) and Loss Density-modulated Prompt Selector (LDS)

# 4 METHOD

## 4.1 OVERVIEW

As discussed in Section 1, the principal bottleneck in prompt-based continual learning is the misselection of prompts for susceptible samples. To address this core issue, we propose a two-fold approach: First, we aim to convert these susceptible samples into resilient ones that are robust to prompt misselection; Second, for the remaining susceptible samples, we focus on enhancing the accuracy of their prompt selection. For formal mathematical definitions and the theoretical foundation that guides our approach, please see Appendix A.

To implement this, we designed a two-stage learning framework as illustrated in Figure 2. It features a Confidence-aware Classifier Alignment (CCA) and a Loss Density-modulated Prompt Selector (LDS), which are specifically designed to address the first and second objectives, respectively. Both stages are built upon our novel sample stratification method, which categorizes samples based on their distinct responses to prompt selection, allowing for this targeted optimization. Similar to prior works (Gao et al., 2024; Wang et al., 2023a), our overall model is composed of a prompt instruction part for learning task-specific prompts, and a prompt selector, which chooses an appropriate prompt for a given sample during inference.

## 4.2 SAMPLE STRATIFICATION BY PROMPT MISSELECTION

In session $t$, given an input-label pair $(x, y)$ to the prompt instruction part, we extract its feature $z = f_\theta(x, P_t)$ under the instruction of the task prompt $P_t$. During classifier alignment, however, $z$ represents a generated feature sampled from a Gaussian distribution. Next, the concatenated classifier $C_{1:t}$ maps the feature vector to the logits corresponding to all classes up to task $t$, i.e., $\mathcal{Y}_{1:t}$: $\ell = C_{1:t}(z)$. The predicted probability for the ground-truth label $y$ is $p_y = \frac{\exp([\ell]_y)}{\sum_{j \in \mathcal{Y}_{1:t}} \exp([\ell]_j)}$, which indicates the confidence level that the model assigns to the sample. The classification loss of the prompt instruction part is $\mathcal{L}_{ce}^{ins} = -\log p_y$.

In Section 1, we conceptually stratify samples into three types based on their responses to prompt misselection. Intuitively, the three types of samples correspond to three distinct levels of confidence to the model, as shown in Figure 1(a). Specifically, resilient samples are consistently classified correctly regardless of which prompt is used. As a result, the model assigns them the highest confidence scores. In contrast, refractory samples are always misclassified no matter the prompt, leading to

the lowest confidence values. Susceptible samples, whose classification outcomes depend on the choice of prompt, exhibit intermediate confidence levels. Moreover, the exponentiated loss $e^{-\mathcal{L}_{\mathrm{ce}}^{\mathrm{ins}}}$, i.e., $p_y$, ranging from low to high, exactly indicates model's increasing confidence. As empirically demonstrated in our preliminary analysis (see Appendix D.3), model confidence, represented by the exponentiated loss, serves as a reliable proxy to distinguish these three sample types. Therefore, we divide the range of $e^{-\mathcal{L}_{\mathrm{ce}}^{\mathrm{ins}}}$ into three intervals, each corresponding to a distinct type of sample. In detail, we set two thresholds $\alpha_1$ and $\alpha_2$, satisfying $0 < \alpha_1 < \alpha_2 < 1$. Samples are categorized as follows: refractory ($< \alpha_1$), susceptible ($\alpha_1 \leq \cdot \leq \alpha_2$), and resilient ($> \alpha_2$). Based on this categorization, we can fully leverage the characteristics of these three types of samples to mitigate task interference caused by prompt misselection.

### 4.3 CONFIDENCE-AWARE CLASSIFIER ALIGNMENT

Since the classifiers are trained independently (as the model only has access to data from the current task at each stage) but work together to determine the decision boundary during testing, many prompt-based methods model old class features with Gaussian distributions and align all classifiers via feature generation (Wang et al., 2023a; Le et al., 2024; Yang et al., 2024). However, this strategy assumes perfect prompt selection because the modeled class features are extracted by their corresponding correct prompts. Consequently, these methods overlook the impact of prompt misselection. To address this issue, our first-stage approach analyzes the distinct impact of the three categories on classifier alignment via feature generation and applies tailored strategies to each of them, thereby transforming susceptible samples into resilient ones during classifier alignment to mitigate task interference caused by prompt misselection.

**Refractory Samples:** The low confidence assigned to refractory samples typically arises from their high similarity to classes in other tasks—a key manifestation of task interference, which is the source of most classification errors in CIL (Nori & Kim, 2024). Focusing on these refractory samples tends to push the decision boundary closer to these classes. As a result, more samples from these classes are pushed near the decision boundary, effectively becoming susceptible samples. During classifier alignment, these newly susceptible samples are regarded as correctly classified, since the features used for Gaussian modeling are extracted using the correct prompt. However, in actual testing, these susceptible samples may cross the decision boundary due to feature drift caused by prompt misselection, leading to misclassification. In essence, the errors caused by prompt misselection can be viewed as a form of feature drift, which only leads to misclassification for susceptible samples that are already close to the decision boundary, as shown in Figure 1(a). Moreover, since refractory samples are inherently ambiguous, prompt selector struggles to assign them the correct prompt. Even if we make them appear to be correctly classified during classifier alignment, they are still likely to be misclassified at test time due to prompt misselection. Based on the analysis above, we directly drop refractory samples during classifier alignment. Unlike previous methods that focus on these samples (Wang et al., 2023a; Le et al., 2024; Yang et al., 2024), dropping them allows many boundary-near susceptible samples to become resilient samples farther from the decision boundary, thereby constructing a prompt-robust decision boundary.

**Resilient Samples:** Resilient samples should be fully leveraged for classifier alignment, as they exhibit highly representative task-specific features and are clearly distinguishable from other tasks. Since each task-specific classifier is trained independently, a cross-task domain gap emerges. Resilient samples, with their representative task features, are best suited to bridge this gap. Therefore, we do not reduce the weight of any resilient sample, as shown in Equation 1.

$$\mathcal{L}_{\mathrm{ca}}(z_{\mathrm{res}}) = \mathcal{L}_{\mathrm{ce}}^{\mathrm{ins}}(z_{\mathrm{res}}). \tag{1}$$

Here, $z_{res}$ denotes generated feature of a resilient sample.

**Susceptible Samples:** Without access to data from previous tasks, it's very difficult for us to learn correct prompt selection for susceptible samples from previous tasks in the current task. As a result, some susceptible samples from previous tasks, especially those that are highly ambiguous, remain challenging for accurate prompt selection. Excessive attention to these samples during classifier alignment thus leads to issues similar to those with refractory samples, where the cost outweighs the benefit. However, as they also capture task-representative knowledge and can be used to refine the decision boundary, dropping them entirely would make the decision boundary too coarse. Therefore, we assign a sinusoidally decaying weight to the relatively difficult susceptible samples to mitigate

their negative impact:

$$\omega(z_{\text{sus}}) = \begin{cases} \lambda[1 - \cos\left(\pi \cdot \frac{l(z_{\text{sus}}) - \alpha_1}{2(\alpha_2 - \alpha_1 - \Delta)}\right)] & \alpha_1 \le l(z_{\text{sus}}) \le \alpha_2 - \Delta \\ 1, & \alpha_2 - \Delta < l(z_{\text{sus}}) < \alpha_2, \end{cases} \quad (2)$$

where $l(z_{sus}) = e^{-\mathcal{L}_{\text{ce}}^{\text{ins}}(z_{sus})}$ denotes the exponentiated loss of generated susceptible sample feature $z_{sus}$, $\Delta \in [0, \alpha_2 - \alpha_1]$ controls the partition point of the piecewise function and $\lambda \in [0, 1]$ is a hyperparameter that controls the degree of down-weighting. Therefore, susceptible samples are utilized to align the classifiers as follows:

$$\mathcal{L}_{\text{ca}}(z_{\text{sus}}) = \text{sg}\left[\omega(z_{\text{sus}})\right] \cdot \mathcal{L}_{\text{ce}}^{\text{ins}}(z_{\text{sus}}). \quad (3)$$

Here $\text{sg}[\cdot]$ denotes the stop-gradient operator as in (Van Den Oord et al., 2017).

By a detailed analysis and targeted utilization of the roles played by the three types of samples, we enable classifiers from different tasks to effectively acquire task-specific knowledge from each other while being robust to prompt misselection and the total loss of Confidence-aware Classifier Alignment is given by $\mathcal{L}_{\text{CCA}} = \mathcal{L}_{\text{ca}}(z_{\text{res}}) + \mathcal{L}_{\text{ca}}(z_{\text{sus}})$.

### 4.4 LOSS DENSITY-MODULATED PROMPT SELECTOR

After obtaining confidence-aware classifier alignment, our second-stage approach focuses on developing a prompt selector that prioritizes accurate prompt prediction for susceptible samples. Our prompt selector is also built on the same frozen backbone $f_\theta$. For each task $t$, we learn a task-specific classifier $S_t$ while keeping all previous classifiers frozen. Additionally, for an input-label pair $(x, y)$, the corresponding confidence score of a classifier $S_k$ is computed as:

$$H_k(x) = \log \sum_{y \in \mathcal{Y}_k} \exp\left(S_k(f_\theta(x))[y]\right) \quad k \in [1, T]. \quad (4)$$

During testing, the task ID is predicted as the index of the classifier with the highest confidence score. Following (Wang et al., 2023b), we use a self-normalization loss to align confidence scores between different classifiers learned from the data stream for meaningful comparison.

**Susceptible Samples:** Our objective of training the prompt selector is to enable correct task ID for susceptible samples. However, susceptible samples constitute only a small fraction of the overall dataset, with the majority of samples being robust. Training only susceptible samples for correct task ID prediction can easily lead to overfitting. Therefore, it is necessary to assign appropriate weights to both susceptible and resilient samples. As shown in Figure 3, the exponentiated loss density of susceptible samples in the intermediate confidence range is significantly lower than that of resilient samples in the high-confidence range. Inspired by (Li et al., 2019), we exploit this disparity in loss density to apply tailored weighting during optimization.

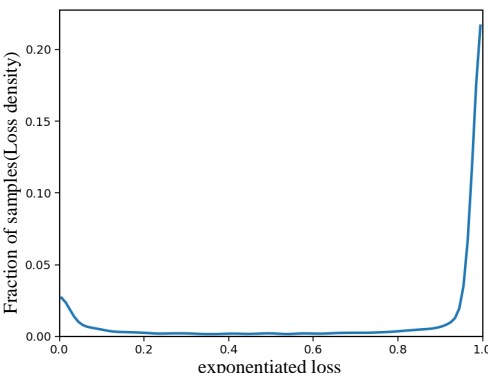

Figure 3: Exponentiated loss distribution

The loss density function of sample $x$ in the current task is defined as:

$$LD(x) = \frac{1}{\epsilon} \sum_{k=1}^{|D_t|} \mathbb{I}\left(|l(x_k) - l(x)| < \frac{\epsilon}{2}\right), \quad (5)$$

where $l(x) = e^{-\mathcal{L}_{\text{ce}}^{\text{ins}}(x)}$ denotes the exponentiated loss of sample $x$, $\epsilon$ is a tiny interval width, and $\mathbb{I}(\cdot)$ is the indicator function, which equals 1 if the condition inside holds and 0 otherwise. $LD(x)$ counts the number of samples whose exponentiated loss falls within an $\epsilon$-sized interval centered at $l(x)$, normalized by the interval width. Then, definition of our modulation factor is formulated as:

$$\Gamma(x) = \frac{|D_t|}{LD(x)}. \quad (6)$$

Based on the modulation factor, we formulate a weighted hinge loss to adaptively up-weight susceptible samples and guide the selector:

$$\mathcal{L}_{ps}(x_{\text{t,sus}}) = \Gamma(x_{\text{t,sus}}) \cdot \left(H_{t-1}^{\max}(x_{\text{t,sus}}) - H_t(x_{\text{t,sus}})\right)_+ \quad (7)$$

$$H_{t-1}^{\max} = \max\{H_i\}, \ i < t. \tag{8}$$

Here, $x_{\text{t,sus}}$ denotes susceptible samples in task $t$ and $(\cdot)_+$ denotes $\max(0, \cdot)$.

**Resilient Samples:** Although resilient samples are insensitive to prompt selection, they can still be leveraged to enhance the acquisition of representative task knowledge. Therefore, we assign them lower weights as in Equation 9, which allows the prompt selector to concentrate on susceptible samples and, at the same time, prevents overfitting.

$$\mathcal{L}_{ps}(x_{\text{t,res}}) = \Gamma(x_{\text{t,res}}) \cdot \left( H_{t-1}^{\max}(x_{\text{t,res}}) - H_t(x_{\text{t,res}}) \right)_+ , \tag{9}$$

where $x_{\text{t,res}}$ denotes resilient samples in task $t$.

**Refractory Samples:** Since we have no access to data from previous tasks, only optimizing susceptible samples as Equation 7 may cause the current classifier to become overly confident, assigning high confidence scores even to data belonging to old tasks. Therefore, we use refractory samples as anchor points to defend such over-confidence:

$$\mathcal{L}_{ps}(x_{\text{t,ref}}) = \left( H_t(x_{\text{t,ref}}) - H_{t-1}^{\max}(x_{\text{t,ref}}) \right)_+ . \tag{10}$$

Here, $x_{\text{t,ref}}$ denotes refractory samples in task $t$. Since refractory samples most resemble old task data compared to the other two categories, we suppress the upper bound confidence on them to alleviate over-confidence.

By employing loss density modulation to focus on susceptible samples and leveraging refractory samples as anchor points, our prompt selector can accurately assign prompts to susceptible samples in new tasks while minimizing the impact on previous tasks and the total loss of Loss Density-modulated Prompt Selector is $\mathcal{L}_{LDS} = \mathcal{L}_{ps}(x_{\text{t,sus}}) + \mathcal{L}_{ps}(x_{\text{t,res}}) + \mathcal{L}_{ps}(x_{\text{t,ref}})$

## 5 EXPERIMENTS

### 5.1 EXPERIMENT SETUP

**Datasets.** We evaluate our approach on four benchmarks under class-incremental learning setting: 10- and 20-split ImageNet-R (Hendrycks et al., 2021), 10-split CIFAR-100 (Krizhevsky et al., 2009) and 10-split DomainNet (Peng et al., 2019). Please refer to Appendix C.1 for more details.

**Baselines and Metrics.** We evaluate our method against state-of-the-art prompt-based methods, including L2P (Wang et al., 2022c), DualPrompt (Wang et al., 2022b), CODA-Prompt (Smith et al., 2023), CPrompt (Gao et al., 2024), and ESN (Wang et al., 2023b), as well as the recent LoRA-based method SD-LoRA (Wu et al., 2025). The upper-bound (UB) performance is achieved by fine-tuning the prompt and classifier with all task data collectively. We evaluate performance using mainly Final Average Accuracy (FAA) and Cumulative Average Accuracy (CAA), which respectively measure the accuracy after the final task and the average accuracy across the entire task sequence (Appendix C.2). It should be noted that some prompt-based methods achieve good performance by implicitly using the task ID during testing, which violates the CIL setting (Appendix D.1). The implementation is described in detail in Appendix C.3.

Table 1: Comparison with existing methods using ImageNet pretrained ViT-B/16. Both the means and standard deviations are reported. * indicates results implemented by (Gao et al., 2024).

| Method | 10-S ImageNet-R | | 20-S ImageNet-R | | 10-S DomainNet | | 10-S CIFAR-100 | |
|---|---|---|---|---|---|---|---|---|
| | FAA (↑) | CAA (↑) | FAA (↑) | CAA (↑) | FAA (↑) | CAA (↑) | FAA (↑) | CAA (↑) |
| UB | 80.27 | — | 80.27 | — | 89.15 | — | 91.99 | — |
| L2P* | $74.6_{\pm 0.90}$ | $80.83_{\pm 1.39}$ | $72.09_{\pm 1.12}$ | $78.39_{\pm 0.94}$ | $81.17_{\pm 0.83}$ | $87.43_{\pm 0.95}$ | $86.38_{\pm 0.31}$ | $91.45_{\pm 0.19}$ |
| DualPrompt* | $74.87_{\pm 0.85}$ | $81.3_{\pm 1.25}$ | $71.69_{\pm 1.06}$ | $79.12_{\pm 1.27}$ | $81.70_{\pm 0.78}$ | $87.80_{\pm 0.99}$ | $86.61_{\pm 0.22}$ | $90.82_{\pm 1.47}$ |
| ESN* | $75.11_{\pm 0.36}$ | $81.63_{\pm 1.10}$ | $70.57_{\pm 0.62}$ | $77.95_{\pm 0.76}$ | $79.22_{\pm 2.04}$ | $86.69_{\pm 1.18}$ | $86.42_{\pm 0.80}$ | $91.65_{\pm 0.67}$ |
| CODA-P* | $75.51_{\pm 0.81}$ | $81.32_{\pm 1.01}$ | $72.25_{\pm 0.78}$ | $78.07_{\pm 0.40}$ | $80.04_{\pm 0.79}$ | $86.27_{\pm 0.82}$ | $85.73_{\pm 0.14}$ | $91.03_{\pm 0.57}$ |
| CPrompt | $77.14_{\pm 0.11}$ | $82.92_{\pm 0.70}$ | $74.79_{\pm 0.28}$ | $81.46_{\pm 0.93}$ | $82.97_{\pm 0.34}$ | $88.54_{\pm 0.41}$ | $87.82_{\pm 0.21}$ | $92.53_{\pm 0.23}$ |
| SD-lora | $77.34_{\pm 0.35}$ | $82.04_{\pm 0.24}$ | $75.26_{\pm 0.37}$ | $80.22_{\pm 0.72}$ | $80.54_{\pm 0.32}$ | $86.68_{\pm 0.06}$ | $88.01_{\pm 0.31}$ | $92.54_{\pm 0.18}$ |
| CoSC | $\mathbf{79.50_{\pm 0.17}}$ | $\mathbf{84.91_{\pm 0.12}}$ | $\mathbf{77.41_{\pm 0.20}}$ | $\mathbf{83.99_{\pm 0.16}}$ | $\mathbf{85.13_{\pm 0.27}}$ | $\mathbf{89.37_{\pm 0.22}}$ | $\mathbf{89.26_{\pm 0.11}}$ | $\mathbf{93.49_{\pm 0.05}}$ |

**Overall Performance:** Table 1 shows the results across multiple runs using the ViT-Base backbone pre-trained with supervised learning on ImageNet. Our method, COSC, demonstrates consistent and superior performance across all benchmarks. Specifically, COSC surpasses prompt-based counterpart, CPrompt in FAA by 2.36% on 10-split ImageNet-R, 2.61% on 20-split ImageNet-R, 2.16% on DomainNet, and 1.44% on CIFAR-100, while also outperforming the LoRA-based method SD-lora.

Table 2: Results on two benchmarks with different self-supervised pre-training paradigms.

| Method | PTM | 10-S ImageNet-R | | 10-S DomainNet | |
|---|---|---|---|---|---|
| | | FAA (↑) | CAA (↑) | FAA (↑) | CAA (↑) |
| CPrompt | | 71.58 | 78.90 | 77.09 | 83.87 |
| SD-lora | iBOT-1K | 69.65 | 66.42 | 74.49 | 71.97 |
| CoSC | | **73.53** | **80.04** | **78.38** | **83.98** |
| CPrompt | | 66.45 | 74.56 | 75.31 | 82.39 |
| SD-lora | DINO-1K | 67.90 | 62.59 | 74.42 | 71.89 |
| CoSC | | **68.80** | **75.59** | **76.82** | **82.41** |

This performance superiority is particularly evident in longer learning sequences: Our advantage over prompt-based methods on ImageNet-R becomes more pronounced in the 20-task split compared to the 10-task split. This trend indicates that COSC effectively mitigates the compounding effects of task interference over time.

To further validate the robustness and generalizability of our approach, we extend our evaluation to backbones pre-trained with self-supervised methods, i.e., iBOT-1K (Zhou et al., 2021) and DINO-1K (Caron et al., 2021). As shown in Table 2, COSC consistently outperforms state-of-the-art prompt-based continual learning methods on these backbones as well, underscoring the generalizability of our strategy and its effectiveness across different pre-training paradigms.

**Ablation Study:** We validate the effectiveness of our two main components, Confidence-aware Classifier Alignment (CCA) and Loss Density-modulated Prompt Selector (LDS). As shown in Table 3, CCA proves crucial for transforming error-prone susceptible samples into robust ones. Its inclusion yields a significant performance boost across all backbones. Notably, this improvement is even more pronounced on self-supervised pre-training paradigms, which demonstrates its strong generalizability and ro-

Table 3: Ablation studies of our CCA and LDS across three pre-training paradigms. FAA results on 10-split ImageNet-R are reported.

| Method | Sup-1K | iBOT-1K | DINO-1K |
|---|---|---|---|
| Baseline | 78.09 | 69.77 | 63.82 |
| w/o CCA | 78.38 | 70.10 | 64.15 |
| w/o LDS | 79.12 | 73.08 | 68.23 |
| **CoSC** | **79.50** | **73.53** | **68.80** |

bustness. The effectiveness of LDS is also clearly demonstrated. Using the supervised paradigm as a test case, the model without LDS alone (w/o LDS) already achieves a high FAA of 79.12%, narrowing the gap to the 80.27% upper bound. Despite this limited room for improvement, our LDS further boosts the performance to 79.50%, closing 33% of the remaining performance gap and validating its ability to refine results even under demanding conditions.

Table 4: Further Analysis of the Effectiveness of CCA. FAA results with supervised paradigm are reported here under the scenario where all samples are assigned to incorrect prompts.

| Method | Split ImageNet-R | Split DomainNet | Split CIFAR-100 |
|---|---|---|---|
| w/o CCA | 70.65 | 78.73 | 83.79 |
| Ours | 76.72 | 82.68 | 87.11 |

**Further Analysis of CCA:** We design CCA to transform susceptible samples into resilient ones, thereby reducing the risk of incorrect predictions caused by prompt misselection. To further evaluate the effectiveness of CCA, we deliberately assigned incorrect prompts to all samples, thus testing its performance under conditions of 100% erroneous prompt selection. As shown in Table 4, our method achieves significant improvement compared to the version without CCA, indicating that CCA effectively transforms susceptible samples into resilient ones. Furthermore, we observe that the improvement in FAA is more pronounced on the most challenging benchmark Split ImageNet-R, with an increase of 6.07%, highlighting the superior ability of CCA to address cross-task interference.

Additionally, to evaluate the effectiveness of CCA under different prompt selector conditions, we replaced our prompt selector with a multi-key matching strategy (Gao et al., 2024) as a simplified version of our approach, i.e., CoSC(simplified). As shown in Figure 4, our CCA consistently improves performance as the number of tasks increases, finally achieving a 2.92% FAA improvement on Split ImageNet-R and a 2.09% improvement on Split CIFAR-100. Moreover, CoSC(simplified)

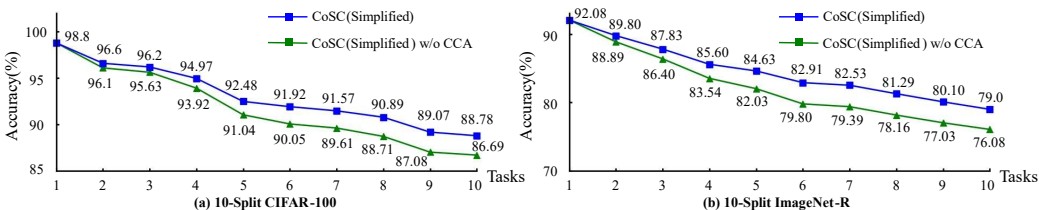

Figure 4: Analysis of the Effectiveness of CCA under a simplified version of our approach, where the prompt selector is replaced with a multi-key matching strategy.

achieves state-of-the-art performance with 79.0% FAA on Split ImageNet-R and 88.78% FAA on Split CIFAR-100 with supervised paradigm, outperforming all comparison methods in Table 1.

Table 5: Analysis of LDS's impact on each sample group. $ACC_T$ represents the total accuracy of the last task. $ACC_{T,res}$, $ACC_{T,sus}$ and $ACC_{T,ref}$ denote the accuracies for resilient, susceptible, and refractory samples in the last task, respectively. Results are from a single run on ImageNet-R.

| Methods | $ACC_{T,res}$ | $ACC_{T,sus}$ | $ACC_{T,ref}$ | $ACC_T$ | FAA |
|---------|---------------|---------------|---------------|---------|-----|
| w/o LDS | 99.57 | 60.34 | 3.85 | 76.99 | 79.23 |
| w/ LDS | 99.78 | 77.59 | 1.54 | 78.22 | 79.67 |

**Further Analysis of LDS:** To further quantify the impact of LDS, we analyze its effect on each sample category in the newest task, as shown in Table 5. The analysis reveals that LDS significantly boosts the accuracy of susceptible samples, and this targeted improvement directly enhances the accuracy of the new task ($ACC_T$). Additionally, the resulting increase in the final FAA gain is also crucial, as it reflects the model's ability to not only improve performance on new tasks but also to resist catastrophic forgetting on old ones, thus creating a more robust and effective prompt selector.

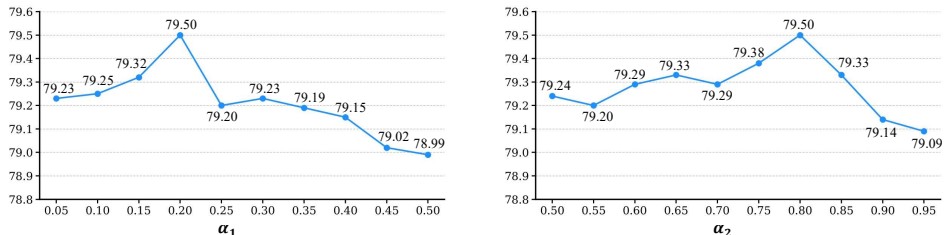

Figure 5: Analysis of the stratification thresholds $\alpha_1$ (left) and $\alpha_2$ (right) on 10-split ImageNet-R.

**Hyperparameter Analysis:** Since our method uses thresholds $\alpha_1$ and $\alpha_2$ for sample stratification, we analyze their impact on FAA performance. To do this, we vary one hyperparameter while keeping the other fixed at its optimal value (i.e., $\alpha_2 = 0.8$ when varying $\alpha_1$, and $\alpha_1 = 0.2$ when varying $\alpha_2$). As shown in Figure 5, the performance curves are relatively flat around their peaks. For instance, the FAA remains high as $\alpha_1$ varies between 0.05 and 0.35, and as $\alpha_2$ ranges from 0.50 to 0.85. This demonstrates that our method is not overly sensitive to the precise choice of $\alpha_1$ and $\alpha_2$, highlighting its robustness. Notably, across this wide range of settings, our method's performance consistently and significantly surpasses all competing methods listed in Table 1.

## 6 CONCLUSION

In this paper, we revisit prompt-based continual learning and find that prompt misselection affects samples in distinct ways. By stratifying samples into susceptible, refractory, and resilient categories, we identify that susceptible samples are the core of the task interference problem. Based on this insight, we propose CoSC, a novel two-stage framework that utilizes Confidence-aware Classifier Alignment (CCA) and a Loss Density-modulated Prompt Selector (LDS) to mitigate interference by specifically targeting these sample types. Extensive experiments across multiple benchmarks and backbones demonstrate that our method achieves state-of-the-art performance under the class-incremental setting, validating the effectiveness and generality of our approach.

## REPRODUCIBILITY STATEMENT

We are committed to ensuring the reproducibility of our research. All experiments were implemented using the PyTorch framework. The implementation details of our proposed CoSC framework are thoroughly described in Appendix C.3. Additionally, our source code is provided in the Supplementary Material. The theoretical foundation of our approach is detailed in the appendices. Appendix A presents the formal definitions of our sample stratification (Definition A.1) and provides a complete proof for the accuracy decomposition in Theorem A.1. Furthermore, the analytical derivation of the theoretically optimal stratification thresholds is provided in Appendix B. The datasets used in our experiments, including ImageNet-R, CIFAR-100, and DomainNet, along with the specific class-incremental splits are detailed in Appendix C.1. The evaluation metrics used to report our results, Final Average Accuracy (FAA) and Cumulative Average Accuracy (CAA), are formally defined in Appendix C.2.

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

# A  THEORETICAL GUIDANCE FOR METHOD DESIGN

This section establishes a theoretical foundation to guide our method's design. We begin by defining the core concepts of our analysis and then decompose the overall prediction accuracy to identify key optimizable factors.

## A.1  PRELIMINARIES AND DEFINITIONS

Let the set of all tasks be $T$. For any input sample $x$ with true label $y$ from task $t \in T$, a prompt-based model first infers a task identity $\hat{t} \in T$ and then makes a prediction using the corresponding task-specific prompt, $P_{\hat{t}}$. We denote the model's final prediction for $x$ when using prompt $P_k$ as $h(x; P_k)$.

Based on the model's behavior with different prompts, we partition the entire data space into three distinct sets of samples, as formally defined below.

**Definition A.1** (Sample Types). *Let $x$ be a sample from task $t$ with true label $y$. The sample $x$ is classified as one of three types:*

- ***Resilient*** *($x \in \mathcal{X}_{res}$): The prediction is correct regardless of the chosen prompt.*

$$h(x; P_k) = y, \quad \forall k \in T$$

- ***Refractory*** *($x \in \mathcal{X}_{ref}$): The prediction is incorrect regardless of the chosen prompt.*

$$h(x; P_k) \neq y, \quad \forall k \in T$$

- ***Susceptible*** *($x \in \mathcal{X}_{sus}$): The prediction is correct with the true task's prompt, but is incorrect with at least one prompt from another task.*

$$h(x; P_t) = y \quad \textit{and} \quad \exists k \in T, k \neq t \text{ s.t. } h(x; P_k) \neq y$$

## A.2  DECOMPOSITION OF PREDICTION ACCURACY

With these definitions, we can decompose the model's overall accuracy.

**Theorem A.1.** *For a prompt-based continual learning model, the overall prediction accuracy $P(y|x)$ for a sample $x$ from task $t$ can be expressed as:*

$$P(y|x) = P(x \in \mathcal{X}_{res}) + P(x \in \mathcal{X}_{sus})\Big[k + (1-k)P(\hat{t} = t | x \in \mathcal{X}_{sus})\Big] \tag{11}$$

*where $k \in [0, 1]$ represents the average accuracy on a susceptible sample given that an incorrect prompt was selected, i.e., $k = P(y|\hat{t} \neq t, x \in \mathcal{X}_{sus})$.*

*Proof.* The proof proceeds in two main steps. First, we decompose the total accuracy by the defined sample types using the law of total probability. Second, we evaluate the conditional accuracy for each sample type.

**Step 1: Decomposition by Sample Type.** The sets $\mathcal{X}_{res}, \mathcal{X}_{sus}, \mathcal{X}_{ref}$ form a partition of the sample space. By the law of total probability, the overall accuracy $P(y|x)$ can be written as:

$$P(y|x) = \sum_{\tau \in \{\text{res,sus,ref}\}} P(y|x \in \mathcal{X}_\tau) \cdot P(x \in \mathcal{X}_\tau) \tag{12}$$

**Step 2: Analysis of Conditional Accuracies.** We now analyze each term $P(y|x \in \mathcal{X}_\tau)$ based on Definition A.1.

**For Resilient Samples** ($x \in \mathcal{X}_{\text{res}}$), the prediction is always correct by definition. Thus,

$$P(y|x \in \mathcal{X}_{\text{res}}) = 1 \tag{13}$$

**For Refractory Samples** ($x \in \mathcal{X}_{\text{ref}}$), the prediction is always incorrect by definition. Thus,

$$P(y|x \in \mathcal{X}_{\text{ref}}) = 0 \tag{14}$$

**For Susceptible Samples** ($x \in \mathcal{X}_{\text{sus}}$), the accuracy depends on whether the correct task prompt is selected. We again use the law of total probability, conditioning on the prompt selection event:

$$
\begin{aligned}
P(y|x \in \mathcal{X}_{\text{sus}}) =& P(y|\hat{t} = t, x \in \mathcal{X}_{\text{sus}})P(\hat{t} = t|x \in \mathcal{X}_{\text{sus}}) \\
& + P(y|\hat{t} \neq t, x \in \mathcal{X}_{\text{sus}})P(\hat{t} \neq t|x \in \mathcal{X}_{\text{sus}})
\end{aligned}
\tag{15}
$$

From Definition A.1, we know that for a susceptible sample, the prediction is correct if the true prompt is used, so $P(y|\hat{t} = t, x \in \mathcal{X}_{\text{sus}}) = 1$. The term $P(y|\hat{t} \neq t, x \in \mathcal{X}_{\text{sus}})$ is the average accuracy given a wrong prompt, which we define as $k$.

Substituting these into Eq. equation 15 and using $P(\hat{t} \neq t|\cdot) = 1 - P(\hat{t} = t|\cdot)$, we get:

$$
\begin{aligned}
P(y|x \in \mathcal{X}_{\text{sus}}) &= 1 \cdot P(\hat{t} = t|x \in \mathcal{X}_{\text{sus}}) + k \cdot \left(1 - P(\hat{t} = t|x \in \mathcal{X}_{\text{sus}})\right) \\
&= P(\hat{t} = t|x \in \mathcal{X}_{\text{sus}}) + k - k \cdot P(\hat{t} = t|x \in \mathcal{X}_{\text{sus}}) \\
&= k + (1 - k)P(\hat{t} = t|x \in \mathcal{X}_{\text{sus}})
\end{aligned}
\tag{16}
$$

**Step 3: Final Combination.** Substituting the conditional accuracies from Eqs. equation 13, equation 14, and equation 16 back into our main decomposition in Eq. equation 12:

$$
\begin{aligned}
P(y|x) &= (1) \cdot P(x \in \mathcal{X}_{\text{res}}) + \left[k + (1 - k)P(\hat{t} = t|x \in \mathcal{X}_{\text{sus}})\right] \cdot P(x \in \mathcal{X}_{\text{sus}}) + (0) \cdot P(x \in \mathcal{X}_{\text{ref}}) \\
&= P(x \in \mathcal{X}_{\text{res}}) + P(x \in \mathcal{X}_{\text{sus}})\left[k + (1 - k)P(\hat{t} = t|x \in \mathcal{X}_{\text{sus}})\right]
\end{aligned}
$$

This completes the proof of Theorem A.1. $\qquad\square$

### A.3 STRATEGIC IMPLICATIONS FOR METHOD DESIGN

This decomposition reveals two primary strategic directions for enhancing model accuracy, which directly motivate the design of our CCA and LDS modules.

**1. The Importance of Sample Type Transformation.** In the accuracy equation, the coefficient for $P(x \in \mathcal{X}_{\text{sus}})$ is $[k + (1 - k)P(\hat{t} = t|x \in \mathcal{X}_{\text{sus}})]$. Since $k < 1$ and $P(\hat{t} = t|\cdot) \leq 1$, this coefficient is strictly less than 1. This proves that resilient samples ($x \in \mathcal{X}_{\text{res}}$) have a greater marginal contribution to the overall accuracy. This finding points to a clear strategy for performance improvement: **transforming a portion of susceptible samples into resilient ones.** This principle is the core motivation behind our CCA module.

**2. The Critical Role of Prompt Selection for Susceptible Samples.** The equation also highlights that the term $P(x \in \mathcal{X}_{\text{sus}})(1 - k)P(\hat{t} = t|x \in \mathcal{X}_{\text{sus}})$ is a key optimizable component. It reveals that for the susceptible set, improving the prompt selection accuracy, $P(\hat{t} = t|x \in \mathcal{X}_{\text{sus}})$, directly enhances final performance. This insight **provides the theoretical impetus for our LDS module**, which is designed specifically to focus on and improve this probability.

## B THEORETICAL DERIVATION OF STRATIFICATION THRESHOLDS

In our method, we employ two thresholds to partition the exponentiated loss into three intervals, thus enabling us to categorize samples into three types. In this section, we analytically derive the two thresholds that meet the necessary criteria and constitute the theoretically optimal solution. Specifically, we show that each interval aligns with a specific sample response to prompt misselection under these two theoretically optimal thresholds, thereby providing theoretical guidance for our experimental design.

Given input sample $x$ with ground-truth label $y$, we define:

- Feature vector under prompt $P_t$: $z = f_\theta(x, P_t) \in \mathbb{R}^d$
- Classifier logits: $\ell = C_{1:t}(z) \in \mathbb{R}^{|\mathcal{Y}_{1:t}|}$
- Exponentiated loss: $l(z) = p_y = \frac{\exp([\ell]_y)}{\sum_{j \in \mathcal{Y}_{1:t}} \exp([\ell]_j)}$

Here, $f_\theta$ denotes the frozen pretrained feature extractor and $C_{1:t}$ is the concatenated classifier of all learned task-specific classifiers.

## B.1 DISTANCE TO DECISION BOUNDARY

First, we need to calculate the distance between the sample features and the decision boundary, as this determines whether the samples may cross the decision boundary under feature shifts induced by prompt misselection.

For a linear classifier $C_{1:t}(z) = Wz + b$, we define the *dominant decision boundary* between the true class $y$ and the set of *conflicting classes* $\mathcal{J} = \{j \mid j \in \{1, \ldots, |\mathcal{Y}_{1:t}|\}, j \neq y\}$, where $\mathcal{Y}_{1:t}$ is the set of all seen classes till task $t$. Let: $j^* = \arg\max_{j \in \mathcal{J}} ([\ell]_j)$ be the *nearest conflicting class*, where $[\ell]_j$ denotes the classifier's output score for class $j$. The signed distance from $z$ to the hyperplane separating $y$ and $j^*$ is:

$$\gamma(z) = \frac{(w_y - w_{j^*})^\top z + (b_y - b_{j^*})}{\|w_y - w_{j^*}\|}. \tag{17}$$

## B.2 RELATIONSHIP BETWEEN CLASSIFICATION LOSS AND DISTANCE TO DECISION BOUNDARY

Next, we need to establish the relationship between the exponentiated classification loss and the distance to the decision boundary, so that the classification loss can be used to measure the sample's response to feature drift caused by prompt misselection.

Let $\Delta_{yj} = [\ell]_y - [\ell]_j = (w_y - w_j)^\top z + (b_y - b_j)$ for any $j \neq y$. The exponentiated classification loss $l(z) = p_y$ satisfies:

$$l(z) = \frac{\exp([\ell]_y)}{\sum_{j \in \mathcal{Y}} \exp([\ell]_j)} = \frac{1}{1 + \sum_{j \neq y} \exp(-\Delta_{yj})}.$$

Thus, the relationship between $l(z)$ and distance to decision boundary $\gamma(z)$ is:

$$l(z) = \frac{1}{1 + e^{-\gamma(z)\|w\|} + C}, \quad \|w\| = \|w_y - w_{j^*}\|, \tag{18}$$

where $C = \sum_{j \neq y, j^*} \exp(-\Delta_{yj})$ quantifies the cumulative effect of non-dominant competitors. From this equation, it can be observed that $l(z)$ is positively correlated with $\gamma(z)$; that is, the larger the value of $l(z)$, the farther the sample feature is from the decision boundary, making it less likely to cross the decision boundary due to feature drift.

## B.3 DERIVATION OF TWO THRESHOLDS

We define the maximum feature drift towards decision boundary caused by prompt misselection as:

$$\delta_{\max} = \max_{t' \neq t} \|f_\theta(x, P_t) - f_\theta(x, P_{t'})\|. \tag{19}$$

Now, we can derive two theoretical thresholds for $l(z)$ based on the numerical value of $\gamma(z)$. $\gamma(z) < 0$ indicates that the sample is always misclassified, regardless of whether prompt misselection occurs. By solving $\gamma(z) = 0$, we obtained the threshold of exponentiated loss that distinguishes susceptible and refractory samples, as shown in Equation 20.

$$\alpha_1 = \sup\{l(z) \mid \gamma(z) = 0, \} = \frac{1}{1 + e^0 + C} = \frac{1}{2 + C}. \tag{20}$$

$\gamma(z) > \delta_{\max}$ indicates that the sample is consistently classified correctly, irrespective of the presence or absence of prompt misselection. By solving $\gamma(z) = \delta_{\max}$, we obtained the threshold of exponentiated loss that distinguishes susceptible and resilient samples, as shown in Equation 21.

$$\alpha_2 = \inf\{l(z) \mid \gamma(z) = \delta_{\max}, \} = \frac{1}{1 + e^{-\|w\|\delta_{\max}} + C}. \tag{21}$$

### B.4 STRATIFICATION BY PROMPT MISSELECTION SENSITIVITY

Based on the above analysis, we identified the theoretically optimal point to categorize the samples into three groups according to their responses to prompt misselection.

**Resilient samples** ($l(z) > \alpha_2$): Implies $\gamma(z) > \delta_{\max}$. *Behavior*: $\forall P_{t'}(t' \neq t), \gamma(z) - \delta_{\max} > 0 \Rightarrow$ predictions stable.

**Susceptible samples** ($\alpha_1 \leq l(z) \leq \alpha_2$): Implies $0 \leq \gamma(z) \leq \delta_{\max}$. *Behavior*: $\exists P_{t'}(t' \neq t)$ s.t. $\gamma(z) - \|\Delta z\| \leq 0 \Rightarrow$ predictions flip.

**Refractory samples** ($l(z) < \alpha_1$): Implies $\gamma(z) < 0$. *Behavior*: $\exists j \neq y$ s.t. $[\ell]_j \geq [\ell]_y \Rightarrow$ inherent misclassification.

## C EXPERIMENTAL SETUPS

### C.1 DATASETS

(i) **ImageNet-R** (Hendrycks et al., 2021) consists of 200 "rendition" categories derived from ImageNet, presenting a significant domain shift due to its diverse artistic styles. It serves as a rigorous benchmark for continual learning. We evaluate performance under two distinct task-division protocols: a standard setup where the 200 classes are randomly partitioned into 10 sequential tasks (20 classes per task), and a more fine-grained setup with 20 sequential tasks (10 classes per task).

(ii) **DomainNet** (Peng et al., 2019) is a large-scale dataset featuring images from six diverse domains. Following the preprocessing protocol of (Gao et al., 2024), we select the 200 categories with the largest number of images and merge samples across all domains. These 200 classes are then randomly partitioned into 10 sequential tasks, with each task comprising 20 classes.

(iii) **CIFAR-100** (Krizhevsky et al., 2009) is a standard benchmark for object recognition, containing 100 distinct classes. For our incremental learning setup, we randomly divide the dataset into 10 sequential tasks, where each new task introduces 10 previously unseen classes.

### C.2 METRICS

**Final Average Accuracy (FAA)** is defined as the average accuracy across all tasks after training on all tasks, which is calculated as:

$$\text{FAA} = \frac{1}{T} \sum_{i=1}^{T} A_{iT},$$

where $T$ is the total number of tasks and $A_{iT}$ is the accuracy on task $i$ after learning task $T$.

**Cumulative Average Accuracy (CAA)** measures the overall incremental performance. Specifically, after learning each task, it computes the average of all previous FAA values up to that point. The calculation is given by:

$$\text{CAA} = \frac{1}{T} \sum_{j=1}^{T} \frac{1}{j} \sum_{i=1}^{j} A_{ij},$$

where $A_{ij}$ denotes the accuracy on task $i$ after learning task $j$.

### C.3 IMPLEMENTATION DETAILS

Given a ViT (Dosovitskiy et al., 2020) comprising an embedding module $f_e$ and a sequence of Transformer blocks $\{f_i\}_{i=1}^{N}$, we proceed as follows: For each session, we generate a task-specific prompt $P_t \in \mathbb{R}^{L_p \times D}$, where $L_p$ is the prompt length and $D$ the hidden size. Rather than appending $P_t$ only at the input layer, we split it into $M$ contiguous segments $\{P_t^{(m)}\}_{m=1}^{M}$, each of size $\frac{L_p}{M} \times D$, and insert $P_t^{(m)}$ into the token sequence immediately before block $f_{i_m}$. This multi-layer insertion enables the prompts to interact with both early and deep representations. During training, only prompt parameters of current task are updated, while keeping all prompts learned from previous tasks frozen.

In our experiments, we divided each task-specific prompt into two segments ($M = 2$) and inserted them into the first and the middle self-attention layers, respectively.

Following prior works (Gao et al., 2024; Smith et al., 2023),we adopt a pre-trained ViT-B/16 as our backbone. We also experiment with two self-supervised VIT-B/16 from iBOT-1K (Zhou et al., 2021) and DINO-1K (Caron et al., 2021). We utilize the SGD optimizer with a momentum of 0.9 and set the initial learning rate to 0.01, which progressively decreased to zero via a cosine annealing schedule. Similar to (Wang et al., 2023a), We perform Gaussian modeling for each category by computing the mean and covariance of features using the corresponding prompt. The training of prompt in the prompt instruction part before our two-stage training approach follows (Gao et al., 2024). Since the frozen pre-trained backbone is limited in its ability to learn our specific optimization objectives for the prompt selector, we learn one continuously trained prompt for the prompt selector and we used the self-normalization loss (Wang et al., 2023b), cross-entropy loss, combined with the proposed LDS to train the prompt selector. It is worth noting that during the testing phase, our method also involves two forward processes, resulting in an inference time cost consistent with existing prompt-based methods. The stratification thresholds hyperparameter $(\alpha_1, \alpha_2)$ are $(0.3, 0.9)$ for CIFAR-100, $(0.2, 0.8)$ for ImageNet-R and DomainNet. $\lambda$ is set to 0.2 for all the benchmarks. The tiny interval width $\epsilon$ is set to 0.005, thus dividing the exponentiated loss into 100 equal-length intervals to calculate loss density in each of them and $\Delta$ that controls the partition point of the piecewise function is set to 0.1. The length of prompts $L_p$ is set to 20. The batch size is set to 16 for the prompt instruction part and 64 for the prompt selector.

# D    MORE EXPERIMENTAL RESULTS

All experimental results reported below are obtained using the ImageNet-pretrained ViT-B/16 model.

## D.1    AN ISSUE WITH EXISTING PROMPT-BASED METHODS

Although some existing prompt-based continual learning methods have been reported to achieve impressive performance, these results rely on the assumption that all input instances within a test batch originate from the same task. Therefore, when task information cannot be utilized (i.e., test batch size = 1), the performance of these methods degrades significantly, as illustrated in Table 6. This violates the Class-Incremental Learning (CIL) setting, which requires that no information about task identity is available for test instances. Under CIL setting, test results should be independent of the test batch size or the order of test instances. We clarify that this constitutes a form of information leakage. Specifically, this issue arises from the incorrect use of the $reshape$ operation. Detailed analysis of this leakage problem can be found in (Feng et al., 2024). In contrast, our method is designed to be free of this information leakage problem, and consequently, its performance is unaffected by the test batch size and remains stable.

RCS-Prompt (Yang et al., 2024) proposes alleviating task interference caused by prompt misselection by relabeling new task samples as old classes. However, as shown in Table 6, RCS-Prompt is highly dependent on the incorrect implementation. Its performance drops significantly when evaluated with a test batch size of 1—where task identity information cannot be exploited. This indicates that RCS-Prompt does not truly address the task interference problem arising from prompt misselection.

Table 6: An implementation issue of existing prompt-based methods. The training procedures follow the official implementations for each method, with only the test batch size being modified. FAA results on three benchmarks are reported.

| Methods | Imagenet-r | | Cifar-100 | | domainnet | |
|---|---|---|---|---|---|---|
| | Test Bs=1 | Test Bs=64 | Test Bs=1 | Test Bs=64 | Test Bs=1 | Test Bs=64 |
| RCS-Prompt (Yang et al., 2024) | 60.53 | 75.07 | 79.42 | 90.50 | 74.81 | 86.67 |
| Hide-Prompt (Wang et al., 2023a) | 65.15 | 75.93 | 83.10 | 90.72 | 77.79 | 86.05 |
| DualPrompt (Wang et al., 2022b) | 69.72 | 70.45 | 85.10 | 86.18 | 68.63 | 73.06 |
| CAPrompt (Li & Zhou, 2025) | 67.10 | 81.48 | 82.68 | 92.80 | 72.66 | 84.49 |
| NoRGa (Le et al., 2024) | 61.30 | 75.65 | 76.90 | 91.21 | 69.96 | 86.75 |

## D.2 FURTHER ABLATION STUDY

Our approach consists of two main modules: Confidence-aware Classifier Alignment (CCA) and Loss Density-modulated Prompt Selector (LDS). Both modules are designed to handle three distinct types of samples separately. To further validate their effectiveness, we conduct ablation studies by individually removing each module's specific treatment for each sample category, thereby demonstrating the necessity of our treatment to each sample category. The experiment is implemented on 10-task Split ImageNet-R.

Table 7: Ablation study on the treatment of each sample category in CCA and LDS, where "w/o resilient", "w/o susceptible", and "w/o refractory" indicate removing our specific treatment for resilient, susceptible, and refractory samples, respectively. FAA on 10-task Split ImageNet-R is reported.

| Method | w/o resilient | w/o susceptible | w/o refractory | Ours |
|--------|---------------|-----------------|----------------|------|
| CCA | 78.27 | 78.93 | 78.57 | 79.50 |
| LDS | 79.25 | 79.12 | 78.72 | 79.50 |

**Further ablation on CCA:** As shown in Table 7, omitting our specific treatment for any sample category results in a notable decrease in performance. Specifically, without resilient samples, cross-task classifiers fail to share representative cross-task knowledge, leading to insufficient knowledge transfer during alignment. Additionally, if we weigh all susceptible samples equally as resilient samples, instead of distinguishing high-confidence and low-confidence subgroups, the classifier may over-shift the decision boundary, causing more resilient samples to become susceptible to prompt misselection. Such over-shift becomes more severe when we pay more attention to refractory samples.

**Further ablation on LDS:** As shown in Table 7, our specific treatment for each sample category consistently yields a gain in overall performance. Without training on resilient samples, the model tends to over-fit to the training set, without acquiring sufficient task knowledge. Without special focus on learning correct prediction for susceptible samples, the prompt selector is not need-oriented and leading many susceptible samples to be misclassified due to prompt misselection. Moreover, without leveraging refractory samples as anchor points, the current classifier may be overly confident, causing sufficient bias towards new tasks, thereby leading to significant catastrophic forgetting.

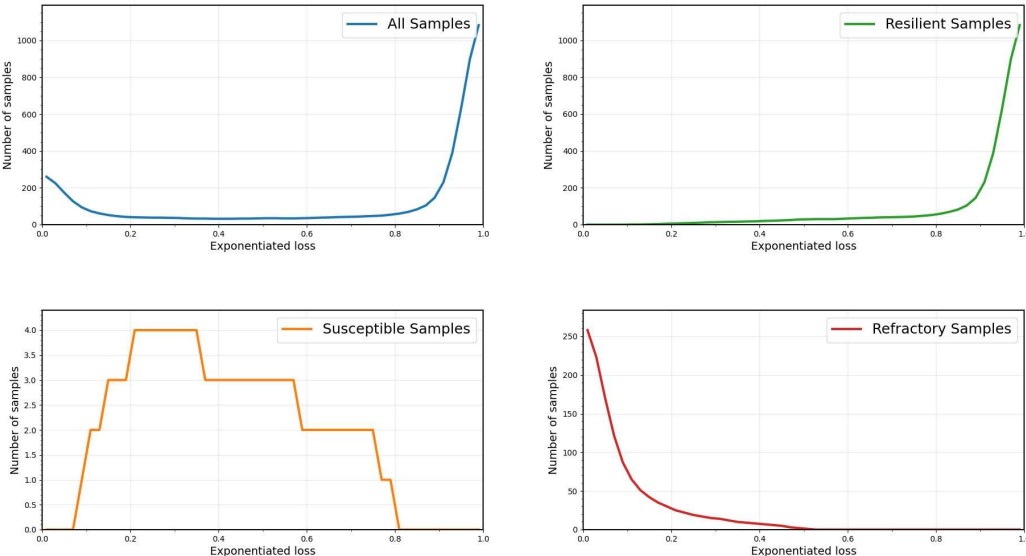

Figure 6: Exponentiated loss distribution of different samples on ImageNet-R.

### D.3 CONFIDENCE-BASED SAMPLE STRATIFICATION

To find an effective way to distinguish between our conceptually defined categories—resilient, susceptible, and refractory—we investigated whether model confidence could serve as a practical separator. To this end, we conducted a preliminary experiment using prompt instruction part of a converged model on 10-task Split ImageNet-R with the corresponding correct prompt for each sample. We first categorized all samples into the three groups based on their actual responses to prompt misselection and then analyzed the distribution of their exponentiated loss, which we use as a proxy for model confidence.

The results, as shown in Figure 6, reveal a key discovery: model confidence serves as a highly effective separator for these groups. It is evident that the categories occupy largely distinct regions of the confidence spectrum: resilient samples are primarily distributed within the interval (0.8, 1), susceptible samples mainly fall within (0.2, 0.8), and refractory samples are concentrated in (0, 0.2). This clear separation demonstrates that a sample's confidence score is a reliable proxy for its sensitivity to prompt misselection. This discovery enables the effective partitioning of samples based on their confidence scores, laying the groundwork for our subsequent approach of applying targeted strategies to each distinct category.

### D.4 A SIMPLIFIED VERSION OF OUR METHOD

To evaluate the performance of our method under different prompt selectors, we replace our prompt selector with a multi-key matching strategy (Gao et al., 2024) as a simplified variant of our approach. We then compare this simplified version with state-of-the-art method of the same type, i.e., CPrompt (Gao et al., 2024) across three datasets. Although the simplified version of our method exhibits a performance gap compared to the full version, it consistently outperforms CPrompt across all three benchmarks, as illustrated in Table 8. This demonstrates the robustness and generalizability of our approach to different prompt selectors with varying levels of accuracy. These results further confirm that our method effectively mitigates task interference caused by prompt misselection at its core.

Table 8: A simplified version of our method. FAA results on three benchmarks are reported.

| Method | Split ImageNet-R | Split DomainNet | Split CIFAR-100 |
|---|---|---|---|
| CPrompt (Gao et al., 2024) | 77.14 | 82.97 | 87.82 |
| Ours(simplified) | 79.00 | 84.38 | 88.68 |
| Ours(full) | 79.50 | 85.13 | 89.26 |

Table 9: FF comparison of our method with state-of-the-art methods. FAA and FF results on three benchmarks are reported.

| Methods | Imagenet-r | | Cifar-100 | | domainnet | |
|---|---|---|---|---|---|---|
| | FAA | FF | FAA | FF | FAA | FF |
| CPrompt (Gao et al., 2024) | 77.14 | 5.97 | 87.82 | 5.06 | 82.97 | 7.45 |
| Ours(simplified) | 79.00 | 4.57 | 88.68 | 4.68 | 84.38 | 6.22 |
| Ours(full) | 79.50 | 4.44 | 89.26 | 4.00 | 85.13 | 4.94 |

### D.5 ANALYSIS OF FORGETTING METRIC

Final Forgetting (FF) quantifies the average performance drop on each task after learning all tasks, reflecting the degree of catastrophic forgetting. It is defined as:

$$\text{FF} = \frac{1}{T-1} \sum_{i=1}^{T-1} \left( \max_{1 \leq j < T} A_{ij} - A_{iT} \right),$$

where $T$ is the total number of tasks, $A_{ij}$ denotes the accuracy on task $i$ after learning task $j$, and $A_{iT}$ is the accuracy on task $i$ after learning all tasks. A higher FF indicates more severe forgetting.

We further compare the Final Forgetting (FF) metric of our method with state-of-the-art approach of the same type across three datasets, as shown in Table 9. Compared to existing methods, our approach not only achieves higher average accuracy but also yields a lower FF score, which indicates superior resistance to catastrophic forgetting. CPrompt also considers the impact of prompt misselection. However, it primarily focuses on its effects within individual tasks. In contrast, our method aims to mitigate cross-task interference caused by prompt misselection, thereby benefiting both new and previous tasks. As a result, our approach not only improves average accuracy but also significantly reduces forgetting.

## LLM USAGE

In the preparation of this manuscript, we utilized a large language model (LLM) as a writing assistance tool. The primary role of the LLM was to polish the text for clarity, conciseness, and grammatical correctness. Specifically, its use was limited to improving sentence structure, correcting spelling and grammar, and rephrasing passages for a more formal academic tone. All scientific contributions, including the core ideas, the development of the proposed methodology, the design and execution of experiments, and the interpretation of results, were exclusively the work of the human authors.

