# OpenReview forum: "In Defense of Prompt-based Continual Learning: Task Interference Mitigation via Confidence-Stratified Classifier Calibration"
_ICLR.cc/2026/Conference — ICLR 2026 Conference Withdrawn Submission_

### Official Review · Reviewer_G1ue · 2025-10-25

**Soundness:** 3
**Presentation:** 2
**Contribution:** 3
**Rating:** 4
**Confidence:** 5

**Summary:**

This paper investigates the problem of task interference in prompt-based continual learning, where misselection of task-specific prompts at inference time substantially undermines model performance. The authors introduce three kinds of data samples, resilient, susceptible, and refractory samples, and show that only susceptible samples are adversely affected by prompt misselection. To address this, the paper proposes a two-stage framework, Confidence-Stratified Classifier Calibration (CoSC), which deploys Confidence-aware Classifier Alignment and a Loss Density-modulated Prompt Selector. Extensive experiments on multiple benchmarks demonstrate that CoSC achieves improved average accuracy and reduced catastrophic forgetting.

**Strengths:**

1. The paper identifies a previously underserved aspect of analyzing prompt misselection in prompt-based continual learning by partitioning data into resilient, susceptible, and refractory groups.
2. Results in Table1 and Table 2 show clear gains over competitive different state-of-the-art methods.

**Weaknesses:**

1. Some symbols in Figure 2 are undefined, which hinders its interpretability. For instance, the meaning of the yellow, red, and pink blocks is unclear.
2. The method relies on exponentiated loss thresholds ($\alpha_1$, $\alpha_2$) to partition data. While Figure 5 argues for robustness to these hyperparameters, the reason for partitioning samples into three categories instead of two or four is not explained.
3. While code is promised and values of some hyperparameters are listed in the Appendices. The setting for building methods in Table 3 is not explained. For instance, it is unclear why the Sup-1k baseline outperforms all proposed methods.
4. The results of 20-S ImageNet-R and 10-S CIFAR-100 with iBOT-1K and DINO-1K are not provided.

**Questions:**

Could the authors explain the performance difference on 10-S DomainNet? The method outperforms CPrompt by 2.16 points under FAA but only by 0.83 under CAA. Providing the per-stage accuracy curves might help elucidate this.

---

### Official Review · Reviewer_xKmK · 2025-10-26

**Soundness:** 3
**Presentation:** 2
**Contribution:** 3
**Rating:** 4
**Confidence:** 4

**Summary:**

The paper defends the effectiveness of prompt-based continual learning (CL) by addressing its main weakness, i.e., task interference caused by prompt misselection. It introduces a new framework, Confidence-Stratified Classifier Calibration (CoSC), which classifies samples into susceptible, refractory, and resilient types based on confidence, targeting the root cause of interference. CoSC combines Confidence-aware Classifier Alignment to make models robust against misselected prompts and a Loss Density-modulated Prompt Selector to improve task identification. Extensive experiments show that this approach significantly enhances accuracy and stability across multiple benchmarks.

**Strengths:**

1. It proposes and empirically validates that prompt misselection has differential impacts across sample types, offering new insight into how task interference arises at the data level.
2. By stratifying samples into three categories, namely, susceptible, refractory, and resilient, based on their distinct responses to prompt misselection, the authors introduce a new perspective that goes beyond uniform treatment of data.
3. Building on this stratification, the paper develops two complementary modules, Confidence-aware Classifier Alignment (CCA) and Loss Density-modulated Prompt Selector (LDS), which effectively leverage the properties of each sample type to mitigate task interference.
4. This paper provides both theoretical justification and empirical support for a new direction in prompt-based continual learning research.

**Weaknesses:**

1. The sample stratification procedure is confusing. The process of obtaining resilient and susceptible samples in Section 4.4 is not sufficiently detailed.

2. Figure 2 lacks visual clarity, particularly in its right panel, where the flow between Confidence-aware Classifier Alignment (CCA) and the Loss Density-modulated Prompt Selector (LDS) is not easily interpretable. The figure should be redesigned to better communicate the two-stage pipeline, explicitly labeling data flow, module inputs/outputs, and how the sample types interact. A clearer diagram would significantly improve accessibility and reader comprehension.

3. The experimental evaluation is focused mainly on standard splits (e.g., 10- and 20-split datasets). To convincingly demonstrate scalability and robustness, results for 5-, 10-, 20-, and 50-split settings are needed. Including these additional results would provide stronger empirical evidence that the proposed method generalizes across varying degrees of task fragmentation and memory constraints, which are central to continual learning scenarios.

**Questions:**

1. The paper lacks clarity on how the Gaussian distribution used for feature generation during classifier alignment is obtained

2. In Section 4.4, the procedure for distinguishing resilient and susceptible samples is ambiguous. It remains unclear whether these samples are directly sampled from the Gaussian distribution, and if so, how different sample types are derived from the same distribution.

3. Figure 2 is visually unclear, especially in the right part, making it difficult for readers to intuitively understand the proposed workflow and conceptual relationships between components.

4. The experimental section would benefit from more extensive results, particularly for 5-, 10-, 20-, and 50-split scenarios, to better evaluate scalability and consistency of performance across varying task numbers.

---

### Official Review · Reviewer_xnDQ · 2025-10-31

**Soundness:** 2
**Presentation:** 2
**Contribution:** 2
**Rating:** 2
**Confidence:** 4

**Summary:**

This paper proposes a new framework, Confidence-Stratified Classifier Calibration (CoSC), to mitigate task interference in prompt-based continual learning. The authors stratify samples into susceptible, refractory, and resilient groups based on their response to prompt misselection, and design two modules—Confidence-aware Classifier Alignment (CCA) and Loss Density-modulated Prompt Selector (LDS)—to improve robustness and prompt selection accuracy. Experiments on several benchmarks show consistent state-of-the-art performance.

**Strengths:**

1. The proposed sample stratification idea is conceptually novel and presents an interesting perspective on analyzing task interference in prompt-based continual learning.

2. The method achieves reasonably strong performance on several standard benchmarks, demonstrating competitive empirical results compared with existing prompt-based baselines.

**Weaknesses:**

## Weaknesses and Questions:
1. **Lack of empirical validation for the proposed three-way sample stratification.**
The method assumes that model confidence can reliably separate samples into susceptible, refractory, and resilient groups, yet this assumption is neither verified nor visualized. There is no experiment demonstrating that the classification aligns with the true behavior of samples under prompt misselection, nor any analysis showing how misclassified samples would affect the learning dynamics.

2. **Inconsistency between theoretical sensitivity and experimental results on threshold parameters.**
Intuitively, the thresholds $\alpha_1$ and $\alpha_2$ are crucial for defining the sample categories. However, the authors report that the method is largely insensitive to these parameters. Such robustness, while seemingly positive, paradoxically undermines the theoretical premise—if performance barely changes with the thresholds, the claimed role of confidence-based stratification may be marginal or even unnecessary.

3. **Unverified assumption that susceptible samples are the sole source of task interference.**
The authors assert that only susceptible samples are influenced by prompt misselection, while refractory and resilient samples remain unaffected. Yet, no experiment directly supports this assumption. There is no ablation where susceptible samples are removed, nor any quantitative evidence that their exclusion or modification alters model performance. The reported ablation study (Table 3) only demonstrates that removing the CCA or LDS modules decreases overall accuracy, which merely confirms that the proposed components are helpful. However, it does not verify that the performance gain stems from correctly identifying or handling susceptible samples, nor does it establish a causal link between sample stratification and the mitigation of task interference.

4. **Questionable necessity of three-way stratification.**
If refractory and resilient samples are indeed insensitive to prompt choice, then treating all data as susceptible—or simply removing the other two categories—should, in theory, yield comparable outcomes. The paper does not present such comparison experiments, making it unclear whether the three-way stratification is truly necessary or merely heuristic. Moreover, this issue echoes the inconsistency noted in the sensitivity analysis (Figure 5): the model’s performance remains largely unaffected by variations in the threshold parameters $\alpha_1$ and $\alpha_2$, which theoretically define the boundaries between these categories. If performance does not depend on how samples are partitioned, it further suggests that the specific stratification itself may be irrelevant, weakening the paper’s central claim that differentiating these sample types is key to mitigating task interference.

5. **Over-idealized theoretical assumption and lack of analysis on approximation error.**
The proposed framework assumes that samples can be cleanly partitioned into three discrete types—susceptible, refractory, and resilient—each exhibiting qualitatively distinct responses to prompt misselection. However, the paper provides no theoretical justification for why such a sharp separation should exist. In practice, both refractory and resilient samples should still be affected by prompt variation to some extent, making their boundaries inherently probabilistic rather than deterministic. The formulation in this paper idealizes these relationships as binary outcomes (“affected” vs. “unaffected”) without analyzing the resulting approximation error or its potential impact on optimization and generalization. As a result, the theoretical model appears over-simplified, lacking discussion of how deviations from the idealized partition would influence the effectiveness or robustness of CoSC.

**Questions:**

See Weaknesses.

---

### Official Review · Reviewer_6hug · 2025-10-31

**Soundness:** 2
**Presentation:** 2
**Contribution:** 3
**Rating:** 4
**Confidence:** 4

**Summary:**

The topic of this paper is about prompt-based incremental learning. The authors propose a method which includes two modules: Confidence-Aware Classifier Alignment (CCF) and Loss Density-Modulated Prompt Selector (LDA). Among them, the CCF module divides samples into three categories by setting a threshold: Resilient, Refractory, and Susceptible samples. Then, different training strategies are set for these three types of samples to train the classification head. The LDA module constructs different training strategies to train the task id classifier by utilizing the Loss Density (LD) between samples, combined with the three categories. The proposed method has been evaluated on multiple tasks.

**Strengths:**

+ Analyzing the Refractory samples and adopting different training strategies for different categories in prompt-based incremental learning is intesting.
+ The performance of the proposed method outperforms that of other methods on multiple experimental settings across multiple datasets.

**Weaknesses:**

- Lack of comparsion to simpler task-id classifiers in previous works (e.g., [a]-[b]). The task-id classifiers based on Gaussian distribution similarity or linear classifiers are utilized in these methods.

[a] Tang L, Tian Z, Li K, et al. Mind the interference: Retaining pre-trained knowledge in parameter efficient continual learning of vision-language models[C]//European conference on computer vision. Cham: Springer Nature Switzerland, 2024: 346-365.

[b] Yu J, Zhuge Y, Zhang L, et al. Boosting continual learning of vision-language models via mixture-of-experts adapters[C]//Proceedings of the IEEE/CVF Conference on Computer Vision and Pattern Recognition. 2024: 23219-23230.
- The extra overhead should be reported. The proposed modules (e.g., CCF and LDA) introduce extra training steps. To show the efficiency of these methods in practical sceneraios, the computational overhead of these modudes should be discussed.
- Some implementation details are unclear: 1) How to select appropriate the hyperparameter of the proposed methods? For instance, the CCF module sets $\alpha_1$ and $\alpha_2$ to 0.8 and 0.2 for ImageNet-R and DomainNet, but to 0.3 and 0.9 for CIFAR100 (refer to the supplementary material). The authors need to make a discussion about the proposed methods’ sensitivity to the hyperparameters. 2) The proposed method divides the samples into three categories: Resilient, Refractory, and Susceptible. The distribution ratios of these categories for different datasets are not discussed in detail.
- The presentation of this paper can be improved. For example, the main pipeline diagram is not clear and the sufficient explanation about this figure are necessary.

**Questions:**

My major concerns are included in the above weaknesses.

---

### Note · Authors · 2025-12-29

I have read and agree with the venue's withdrawal policy on behalf of myself and my co-authors.